# Bilinear Attention Networks

**Jin-Hwa Kim**[1]*, **Jaehyun Jun**[2], **Byoung-Tak Zhang**[2,3]
[1]SK T-Brain, [2]Seoul National University, [3]Surromind Robotics
jnhwkim@sktbrain.com, {jhjun,btzhang}@bi.snu.ac.kr

## Abstract

Attention networks in multimodal learning provide an efficient way to utilize given visual information selectively. However, the computational cost to learn attention distributions for every pair of multimodal input channels is prohibitively expensive. To solve this problem, co-attention builds two separate attention distributions for each modality neglecting the interaction between multimodal inputs. In this paper, we propose bilinear attention networks (BAN) that find bilinear attention distributions to utilize given vision-language information seamlessly. BAN considers bilinear interactions among two groups of input channels, while low-rank bilinear pooling extracts the joint representations for each pair of channels. Furthermore, we propose a variant of multimodal residual networks to exploit eight-attention maps of the BAN efficiently. We quantitatively and qualitatively evaluate our model on visual question answering (VQA 2.0) and Flickr30k Entities datasets, showing that BAN significantly outperforms previous methods and achieves new state-of-the-arts on both datasets.

## 1   Introduction

Machine learning for computer vision and natural language processing accelerates the advancement of artificial intelligence. Since vision and natural language are the major modalities of human interaction, understanding and reasoning of vision and natural language information become a key challenge. For instance, visual question answering involves a vision-language cross-grounding problem. A machine is expected to answer given questions like *"who is wearing glasses?"*, *"is the umbrella upside down?"*, or *"how many children are in the bed?"* exploiting visually-grounded information.

For this reason, visual attention based models have succeeded in multimodal learning tasks, identifying selective regions in a spatial map of an image defined by the model. Also, textual attention can be considered along with visual attention. The attention mechanism of co-attention networks [36, 18, 20, 39] concurrently infers visual and textual attention distributions for each modality. The co-attention networks selectively attend to question words in addition to a part of image regions. However, the co-attention neglects the interaction between words and visual regions to avoid increasing computational complexity.

In this paper, we extend the idea of co-attention into bilinear attention which considers every pair of multimodal channels, *e.g.*, the pairs of question words and image regions. If the given question involves multiple visual concepts represented by multiple words, the inference using visual attention distributions for each word can exploit relevant information better than that using single compressed attention distribution.

From this background, we propose bilinear attention networks (BAN) to use a bilinear attention distribution, on top of low-rank bilinear pooling [15]. Notice that the BAN exploits bilinear interactions between two groups of input channels, while low-rank bilinear pooling extracts the joint representations for each pair of channels. Furthermore, we propose a variant of multimodal residual

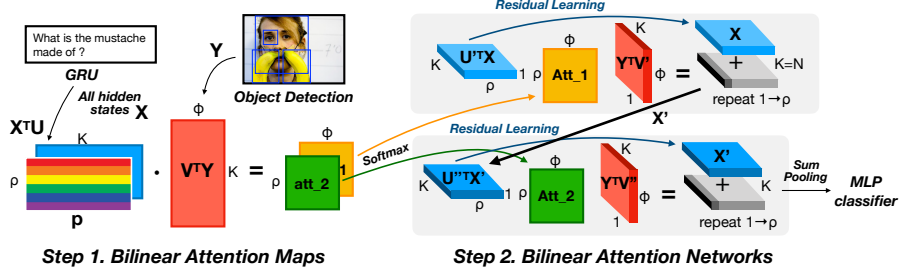

Figure 1: Overview of the two-glimpse BAN. Two multi-channel inputs, $\phi$-object detection features and $\rho$-length GRU hidden vectors, are used to get bilinear attention maps and joint representations to be used by a classifier. For the definition of the BAN, see the text in Section 3.

networks (MRN) to efficiently utilize the multiple bilinear attention maps of the BAN, unlike the previous works [6, 15] where multiple attention maps are used by concatenating the attended features. Since the proposed residual learning method for BAN exploits residual summations instead of concatenation, which leads to parameter-efficiently and performance-effectively learn up to eight-glimpse BAN. For the overview of the two-glimpse BAN, please refer to Figure 1.

Our main contributions are:

- We propose the bilinear attention networks (BAN) to learn and use bilinear attention distributions, on top of the low-rank bilinear pooling technique.

- We propose a variant of multimodal residual networks (MRN) to efficiently utilize the multiple bilinear attention maps generated by our model. Unlike previous works, our method successfully utilizes up to 8 attention maps.

- Finally, we validate our proposed method on a large and highly-competitive dataset, VQA 2.0 [8]. Our model achieves a new state-of-the-art maintaining simplicity of model structure. Moreover, we evaluate the visual grounding of bilinear attention map on Flickr30k Entities [23] outperforming previous methods, along with 25.37% improvement of inference speed taking advantage of the processing of multi-channel inputs.

## 2   Low-rank bilinear pooling

We first review the low-rank bilinear pooling and its application to attention networks [15], which uses single-channel input (question vector) to combine the other multi-channel input (image features) as single-channel intermediate representation (attended feature).

**Low-rank bilinear model.** The previous works [35, 22] proposed a low-rank bilinear model to reduce the rank of bilinear weight matrix $\mathbf{W}_i$ to give regularity. For this, $\mathbf{W}_i$ is replaced with the multiplication of two smaller matrices $\mathbf{U}_i\mathbf{V}_i^T$, where $\mathbf{U}_i \in \mathbb{R}^{N \times d}$ and $\mathbf{V}_i \in \mathbb{R}^{M \times d}$. As a result, this replacement makes the rank of $\mathbf{W}_i$ to be at most $d \leq \min(N, M)$. For the scalar output $f_i$ (bias terms are omitted without loss of generality):

$$f_i = \mathbf{x}^T\mathbf{W}_i\mathbf{y} \approx \mathbf{x}^T\mathbf{U}_i\mathbf{V}_i^T\mathbf{y} = \mathbb{1}^T(\mathbf{U}_i^T\mathbf{x} \circ \mathbf{V}_i^T\mathbf{y}) \tag{1}$$

where $\mathbb{1} \in \mathbb{R}^d$ is a vector of ones and $\circ$ denotes Hadamard product (element-wise multiplication).

**Low-rank bilinear pooling.** For a vector output $\mathbf{f}$, a pooling matrix $\mathbf{P}$ is introduced:

$$\mathbf{f} = \mathbf{P}^T(\mathbf{U}^T\mathbf{x} \circ \mathbf{V}^T\mathbf{y}) \tag{2}$$

where $\mathbf{P} \in \mathbb{R}^{d \times c}$, $\mathbf{U} \in \mathbb{R}^{N \times d}$, and $\mathbf{V} \in \mathbb{R}^{M \times d}$. It allows $\mathbf{U}$ and $\mathbf{V}$ to be two-dimensional tensors by introducing $\mathbf{P}$ for a vector output $\mathbf{f} \in \mathbb{R}^c$, significantly reducing the number of parameters.

**Unitary attention networks.** Attention provides an efficient mechanism to reduce input channel by selectively utilizing given information. Assuming that a multi-channel input $\mathbf{Y}$ consisting of $\phi = |\{\mathbf{y}_i\}|$ column vectors, we want to get single channel $\hat{\mathbf{y}}$ from $\mathbf{Y}$ using the weights $\{\alpha_i\}$:

$$\hat{\mathbf{y}} = \sum_i \alpha_i\mathbf{y}_i \tag{3}$$

where $\alpha$ represents an attention distribution to selectively combine $\phi$ input channels. Using the low-rank bilinear pooling, the $\alpha$ is defined by the output of softmax function as:

$$\alpha := \text{softmax}\Big(\mathbf{P}^T\big((\mathbf{U}^T\mathbf{x} \cdot \mathbb{1}^T) \circ (\mathbf{V}^T\mathbf{Y})\big)\Big) \tag{4}$$

where $\alpha \in \mathbb{R}^{G \times \phi}$, $\mathbf{P} \in \mathbb{R}^{d \times G}$, $\mathbf{U} \in \mathbb{R}^{N \times d}$, $\mathbf{x} \in \mathbb{R}^N$, $\mathbb{1} \in \mathbb{R}^{\phi}$, $\mathbf{V} \in \mathbb{R}^{M \times d}$, and $\mathbf{Y} \in \mathbb{R}^{M \times \phi}$. If $G > 1$, multiple glimpses (*a.k.a.* attention heads) are used [13, 6, 15], then $\hat{\mathbf{y}} = \|_{g=1}^{G} \sum_i \alpha_{g,i}\mathbf{y}_i$, the concatenation of attended outputs. Finally, two single channel inputs $\mathbf{x}$ and $\hat{\mathbf{y}}$ can be used to get the joint representation using the other low-rank bilinear pooling for a classifier.

# 3  Bilinear attention networks

We generalize a bilinear model for two multi-channel inputs, $\mathbf{X} \in \mathbb{R}^{N \times \rho}$ and $\mathbf{Y} \in \mathbb{R}^{M \times \phi}$, where $\rho = |\{\mathbf{x}_i\}|$ and $\phi = |\{\mathbf{y}_j\}|$, the numbers of two input channels, respectively. To reduce both input channel simultaneously, we introduce bilinear attention map $\mathcal{A} \in \mathbb{R}^{\rho \times \phi}$ as follows:

$$\mathbf{f}'_k = (\mathbf{X}^T\mathbf{U}')_k^T \mathcal{A}(\mathbf{Y}^T\mathbf{V}')_k \tag{5}$$

where $\mathbf{U}' \in \mathbb{R}^{N \times K}$, $\mathbf{V}' \in \mathbb{R}^{M \times K}$, $(\mathbf{X}^T\mathbf{U}')_k \in \mathbb{R}^{\rho}$, $(\mathbf{Y}^T\mathbf{V}')_k \in \mathbb{R}^{\phi}$, and $\mathbf{f}'_k$ denotes the $k$-th element of intermediate representation. The subscript $k$ for the matrices indicates the index of column. Notice that Equation 5 is a bilinear model for the two groups of input channels where $\mathcal{A}$ in the middle is a bilinear weight matrix. Interestingly, Equation 5 can be rewritten as:

$$\mathbf{f}'_k = \sum_{i=1}^{\rho}\sum_{j=1}^{\phi} \mathcal{A}_{i,j}(\mathbf{X}_i^T\mathbf{U}'_k)(\mathbf{V}'^T_k\mathbf{Y}_j) = \sum_{i=1}^{\rho}\sum_{j=1}^{\phi} \mathcal{A}_{i,j}\mathbf{X}_i^T(\mathbf{U}'_k\mathbf{V}'^T_k)\mathbf{Y}_j \tag{6}$$

where $\mathbf{X}_i$ and $\mathbf{Y}_j$ denotes the $i$-th channel (column) of input $\mathbf{X}$ and the $j$-th channel (channel) of input $\mathbf{Y}$, respectively, $\mathbf{U}'_k$ and $\mathbf{V}'_k$ denotes the $k$-th column of $\mathbf{U}'$ and $\mathbf{V}'$ matrices, respectively, and $\mathcal{A}_{i,j}$ denotes an element in the $i$-th row and the $j$-th column of $\mathcal{A}$. Notice that, for each pair of channels, the 1-rank bilinear representation of two feature vectors is modeled in $\mathbf{X}_i^T(\mathbf{U}'_k\mathbf{V}'^T_k)\mathbf{Y}_j$ of Equation 6 (eventually at most $K$-rank bilinear pooling for $\mathbf{f}' \in \mathbb{R}^K$). Then, the bilinear joint representation is $\mathbf{f} = \mathbf{P}^T\mathbf{f}'$ where $\mathbf{f} \in \mathbb{R}^C$ and $\mathbf{P} \in \mathbb{R}^{K \times C}$. For the convenience, we define the bilinear attention networks as a function of two multi-channel inputs parameterized by a bilinear attention map as follows:

$$\mathbf{f} = \text{BAN}(\mathbf{X}, \mathbf{Y}; \mathcal{A}). \tag{7}$$

**Bilinear attention map.** Now, we want to get the attention map similarly to Equation 4. Using Hadamard product and matrix-matrix multiplication, the attention map $\mathcal{A}$ is defined as:

$$\mathcal{A} := \text{softmax}\Big(\big((\mathbb{1} \cdot \mathbf{p}^T) \circ \mathbf{X}^T\mathbf{U}\big)\mathbf{V}^T\mathbf{Y}\Big) \tag{8}$$

where $\mathbb{1} \in \mathbb{R}^{\rho}$, $\mathbf{p} \in \mathbb{R}^{K'}$, and remind that $\mathcal{A} \in \mathbb{R}^{\rho \times \phi}$. The softmax function is applied element-wisely. Notice that each logit $A_{i,j}$ of the softmax is the output of low-rank bilinear pooling as:

$$A_{i,j} = \mathbf{p}^T\big((\mathbf{U}^T\mathbf{X}_i) \circ (\mathbf{V}^T\mathbf{Y}_j)\big). \tag{9}$$

The multiple bilinear attention maps can be extended as follows:

$$\mathcal{A}_g := \text{softmax}\Big(\big((\mathbb{1} \cdot \mathbf{p}_g^T) \circ \mathbf{X}^T\mathbf{U}\big)\mathbf{V}^T\mathbf{Y}\Big) \tag{10}$$

where the parameters of $\mathbf{U}$ and $\mathbf{V}$ are shared, but not for $\mathbf{p}_g$ where $g$ denotes the index of glimpses.

**Residual learning of attention.** Inspired by multimodal residual networks (MRN) from Kim et al. [14], we propose a variant of MRN to integrate the joint representations from the multiple bilinear attention maps. The $i + 1$-th output is defined as:

$$\mathbf{f}_{i+1} = \text{BAN}_i(\mathbf{f}_i, \mathbf{Y}; \mathcal{A}_i) \cdot \mathbb{1}^T + \mathbf{f}_i \tag{11}$$

where $\mathbf{f}_0 = \mathbf{X}$ (if $N = K$) and $\mathbb{1} \in \mathbb{R}^{\rho}$. Here, the size of $\mathbf{f}_i$ is the same with the size of $\mathbf{X}$ as successive attention maps are processed. To get the logits for a classifier, *e.g.*, two-layer MLP, we sum over the channel dimension of the last output $\mathbf{f}_G$, where $G$ is the number of glimpses.

**Time complexity.** When we assume that the number of input channels is smaller than feature sizes, $M \geq N \geq K \gg \phi \geq \rho$, the time complexity of the BAN is the same with the case of one multi-channel input as $\mathcal{O}(KM\phi)$ for single glimpse model. Since the BAN consists of matrix chain multiplication and exploits the property of low-rank factorization in the low-rank bilinear pooling.

# 4  Related works

**Multimodal factorized bilinear pooling.** Yu et al. [39] extends low-rank bilinear pooling [15] using the rank > 1. They remove a projection matrix $\mathbf{P}$, instead, $d$ in Equation 2 is replaced with much smaller $k$ while $\mathbf{U}$ and $\mathbf{V}$ are three-dimensional tensors. However, this generalization was not effective for BAN, at least in our experimental setting. Please see *BAN-1+MFB* in Figure 2b where the performance is not significantly improved from that of *BAN-1*. Furthermore, the peak GPU memory consumption is larger due to its model structure which hinders to use multiple-glimpse BAN.

**Co-attention networks.** Xu and Saenko [36] proposed the spatial memory network model estimating the correlation among every image patches and tokens in a sentence. The estimated correlation $\mathbf{C}$ is defined as $(\mathbf{U}\mathbf{X})^T\mathbf{Y}$ in our notation. Unlike our method, they get an attention distribution $\alpha = \mathrm{softmax}\left( \max_{i=1,\ldots,\rho}(\mathbf{C}_i) \right) \in \mathbb{R}^\rho$ where the logits to $\mathrm{softmax}$ are the maximum values in each row vector of $\mathbf{C}$. The attention distribution for the other input can be calculated similarly. There are variants of co-attention networks [18, 20], especially, Lu et al. [18] sequentially get two attention distributions conditioning on the other modality. Recently, Yu et al. [39] reduce the co-attention method into two steps, self-attention for a question embedding and the question-conditioned attention for a visual embedding. However, these co-attention approaches use separate attention distributions for each modality, neglecting the interaction between the modalities what we consider and model.

# 5  Experiments

## 5.1  Datasets

**Visual Question Answering (VQA).** We evaluate on the VQA 2.0 dataset [1, 8], which is improved from the previous version to emphasize visual understanding by reducing the answer bias in the dataset. This improvement pushes the model to have the more effective joint representation of question and image, which fits the motivation of our bilinear attention approach. The VQA evaluation metric considers inter-human variability defined as $\mathrm{Accuracy}(ans) = \min(\#\text{humans that said } ans/3, 1)$. Note that reporting accuracies are averaged over all ten choose nine sets of ground-truths. The test set is split into test-dev, test-standard, test-challenge, and test-reserve. The annotations for the test set are unavailable except the remote evaluation servers.

**Flickr30k Entities.** For the evaluation of visual grounding by the bilinear attention maps, we use Flickr30k Entities [23] consisting of 31,783 images [38] and 244,035 annotations that multiple entities (phrases) in a sentence for an image are mapped to the boxes on the image to indicate the correspondences between them. The task is to localize a corresponding box for each entity. In this way, visual grounding of textual information is quantitatively measured. Following the evaluation metric [23], if a predicted box has the intersection over union (IoU) of overlapping area with one of the ground-truth boxes which are greater than or equal to 0.5, the prediction for a given entity is correct. This metric is called Recall@1. If K predictions are permitted to find at least one correction, it is called Recall@K. We report Recall@1, 5, and 10 to compare state-of-the-arts (R@K in Table 4). The upper bound of performance depends on the performance of object detection if the detector proposes candidate boxes for the prediction.

## 5.2  Preprocessing

**Question embedding.** For VQA, we get a question embedding $\mathbf{X}^T \in \mathbb{R}^{14 \times N}$ using GloVe word embeddings [21] and the outputs of Gated Recurrent Unit (GRU) [5] for every time-steps up to the first 14 tokens following the previous work [29]. The questions shorter than 14 words are end-padded with zero vectors. For Flickr30k Entities, we use a full length of sentences (82 is maximum) to get all entities. We mark the token positions which are at the end of each annotated phrase. Then, we select a subset of the output channels of GRU using these positions, which makes the number of channels is the number of entities in a sentence. The word embeddings and GRU are fine-tuned in training.

**Image features.** We use the image features extracted from bottom-up attention [2]. These features are the output of Faster R-CNN [25], pre-trained using Visual Genome [17]. We set a threshold for object detection to get $\phi = 10$ to 100 objects per image. The features are represented as $\mathbf{Y}^T \in \mathbb{R}^{\phi \times 2,048}$, which is fixed while training. To deal with variable-channel inputs, we mask the padding logits with minus infinite to get zero probability from $\mathrm{softmax}$ avoiding underflow.

## 5.3 Nonlinearity and classifier

**Nonlinearity.** We use ReLU [19] to give nonlinearity to BAN:

$$\mathbf{f}'_k = \sigma(\mathbf{X}^T\mathbf{U}')^T_k \cdot \mathcal{A} \cdot \sigma(\mathbf{Y}^T\mathbf{V}')_k \tag{12}$$

where $\sigma$ denotes $\text{ReLU}(x) := \max(x, 0)$. For the attention maps, the logits are defined as:

$$A := \left((\mathbb{1} \cdot \mathbf{p}^T) \circ \sigma(\mathbf{X}^T\mathbf{U})\right) \cdot \sigma(\mathbf{V}^T\mathbf{Y}). \tag{13}$$

**Classifier.** For VQA, we use a two-layer multi-layer perceptron as a classifier for the final joint representation $\mathbf{f}_G$. The activation function is ReLU. The number of outputs is determined by the minimum occurrence of the answer in unique questions as nine times in the dataset, which is 3,129. Binary cross entropy is used for the loss function following the previous work [29]. For Flickr30k Entities, we take the output of bilinear attention map, and binary cross entropy is used for this output.

## 5.4 Hyperparameters and regularization

**Hyperparameters.** The size of image features and question embeddings are $M = 2,048$ and $N = 1,024$, respectively. The size of joint representation $C$ is the same with the rank $K$ in low-rank bilinear pooling, $C = K = 1,024$, but $K' = K \times 3$ is used in the bilinear attention maps to increase a representational capacity for residual learning of attention. Every linear mapping is regularized by Weight Normalization [27] and Dropout [28] ($p = .2$, except for the classifier with .5). Adamax optimizer [16], a variant of Adam based on infinite norm, is used. The learning rate is $\min(ie^{-3}, 4e^{-3})$ where $i$ is the number of epochs starting from 1, then after 10 epochs, the learning rate is decayed by 1/4 for every 2 epochs up to 13 epochs (*i.e.* $1e^{-3}$ for 11-th and $2.5e^{-4}$ for 13-th epoch). We clip the 2-norm of vectorized gradients to .25. The batch size is 512.

**Regularization.** For the test split of VQA, both train and validation splits are used for training. We augment a subset of Visual Genome [17] dataset following the procedure of the previous works [29]. Accordingly, we adjust the model capacity by increasing all of $N$, $C$, and $K$ to 1,280. And, $G = 8$ glimpses are used. For Flickr30k Entities, we use the same test split of the previous methods [23], without additional hyperparameter tuning from VQA experiments.

## 6 VQA results and discussions

### 6.1 Quantitative results

**Comparison with state-of-the-arts.** The first row in Table 1 shows 2017 VQA Challenge winner architecture [2, 29]. BAN significantly outperforms this baseline and successfully utilize up to eight bilinear attention maps to improve its performance taking advantage of residual learning of attention. As shown in Table 3, BAN outperforms the latest model [39] which uses the same bottom-up attention feature [2] by a substantial margin. *BAN-Glove* uses the concatenation of 300-dimensional Glove word embeddings and the semantically-closed mixture of these embeddings (see Appendix A.1). Notice that similar approaches can be found in the competitive models [6, 39] in Table 3 with a different initialization strategy for the same 600-dimensional word embedding. *BAN-Glove-Counter* uses both the previous 600-dimensional word embeddings and counting module [41], which exploits spatial information of detected object boxes from the feature extractor [2]. The learned representation $\mathbf{c} \in \mathbb{R}^{\phi+1}$ for the counting mechanism is linearly projected and added to the joint representation after applying ReLU (see Equation 15 in Appendix A.2). In Table 5 (Appendix), we compare with the entries in the leaderboard of both VQA Challenge 2017 and 2018 achieving the 1st place at the time of submission (our entry is not shown in the leaderboard since challenge entries are not visible).

**Comparison with other attention methods.** *Unitary attention* has a similar architecture with Kim et al. [15] where a question embedding vector is used to calculate the attentional weights for multiple image features of an image. *Co-attention* has the same mechanism of Yu et al. [39], similar to Lu et al. [18], Xu and Saenko [36], where multiple question embeddings are combined as single embedding vector using a self-attention mechanism, then unitary visual attention is applied. Table 2 confirms that bilinear attention is significantly better than any other attention methods. The co-attention is slightly better than simple unitary attention. In Figure 2a, co-attention suffers overfitting more severely (green) than any other methods, while bilinear attention (blue) is more regularized compared with the

Table 1: Validation scores on VQA 2.0 dataset for the number of glimpses of the BAN. The standard deviations are reported after ± using three random initialization.

| Model | VQA Score |
|---|---|
| Bottom-Up [29] | 63.37 ±0.21 |
| BAN-1 | |
| BAN-2 | |
| BAN-4 | |
| BAN-8 | |
| BAN-12 | 66.04 ±0.08 |

Table 2: Validation scores on VQA 2.0 dataset for attention and integration mechanisms. The *nParams* indicates the number of parameters. Note that the hidden sizes of unitary attention and co-attention are 1,280, while 1,024 for the BAN.

| Model | nParams | VQA Score |
|---|---|---|
| Unitary attention | 31.9M | 64.59 ±0.04 |
| Co-attention | 32.5M | 64.79 ±0.06 |
| Bilinear attention | 32.2M | **65.36 ±0.14** |
| BAN-4 | 44.8M | **65.81 ±0.09** |
| BAN-4 (sum) | 44.8M | 64.78 ±0.08 |
| BAN-4 (concat) | 51.1M | 64.71 ±0.21 |

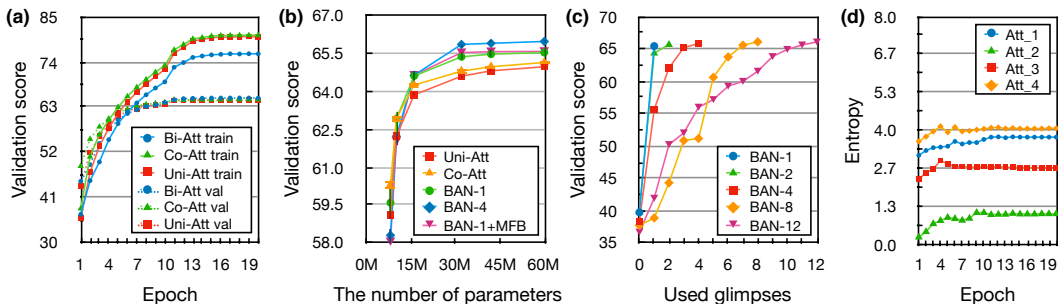

Figure 2: **(a)** learning curves. Bilinear attention (bi-att) is more robust to overfitting than unitary attention (uni-att) and co-attention (co-att). **(b)** validation scores for the number of parameters. The error bar indicates the standard deviation among three random initialized models, although it is too small to be noticed for over-15M parameters. **(c)** ablation study for the first-N-glimpses (x-axis) used in the evaluation. **(d)** the information entropy (y-axis) for each attention map in the four-glimpse BAN. The entropy of multiple attention maps is converged to certain levels.

others. In Figure 2b, BAN is the most parameter-efficient among various attention methods. Notice that four-glimpse BAN more parsimoniously utilizes its parameters than one-glimpse BAN does.

## 6.2 Residual learning of attention

**Comparison with other approaches.** In the second section of Table 2, the residual learning of attention significantly outperforms the other methods, *sum, i.e.,* $\mathbf{f}_G = \sum_i \mathrm{BAN}_i(\mathbf{X}, \mathbf{Y}; \mathcal{A}_i)$, and *concatenation (concat), i.e.,* $\mathbf{f}_G = \|_i \mathrm{BAN}_i(\mathbf{X}, \mathbf{Y}; \mathcal{A}_i)$. Whereas, the difference between *sum* and *concat* is not significantly different. Notice that the number of parameters of *concat* is larger than the others since the input size of the classifier is increased.

**Ablation study.** An interesting property of residual learning is robustness toward arbitrary ablations [31]. To see the relative contributions, we observe the learning curve of validation scores when incremental ablation is performed. First, we train {1,2,4,8,12}-glimpse models using training split. Then, we evaluate the model on validation split using the first $N$ attention maps. Hence, the intermediate representation $\mathbf{f}_N$ is directly fed into the classifier instead of $\mathbf{f}_G$. As shown in Figure 2c, the accuracy gain of the first glimpse is the highest, then the gain is smoothly decreased as the number of used glimpses is increased.

**Entropy of attention.** We analyze the information entropy of attention distributions in a four-glimpse BAN. As shown in Figure 2d, the mean entropy of each attention for validation split is converged to a different level of values. This result is repeatably observed in the other number of glimpse models. Our speculation is the multi-attention maps do not equally contribute similarly to voting by committees, but the residual learning by the multi-step attention. We argue that this is a novel observation where the residual learning [9] is used for stacked attention networks.

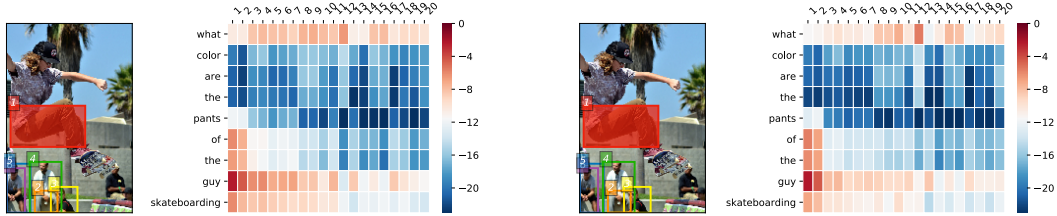

Figure 3: Visualization of the bilinear attention maps for two-glimpse BAN. The left and right groups indicate the first and second bilinear attention maps (right in each group, log-scaled) and the visualized image (left in each group). The most salient six boxes (1-6 numbered in the images and x-axis of the grids) in the first attention map determined by marginalization are visualized on both images to compare. The model gives the correct answer, *brown*.

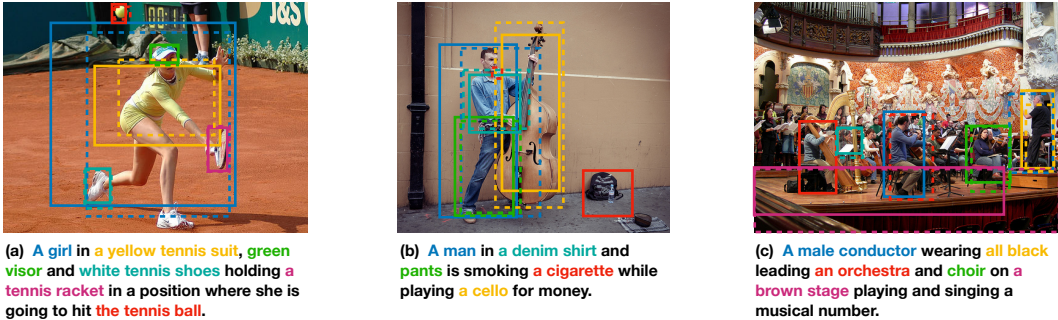

(a) **A girl** in **a yellow tennis suit**, **green visor** and **white tennis shoes** holding **a tennis racket** in a position where she is going to hit **the tennis ball**.

(b) **A man** in **a denim shirt** and **pants** is smoking **a cigarette** while playing **a cello** for money.

(c) **A male conductor** wearing **all black** leading **an orchestra** and **choir** on **a brown stage** playing and singing a musical number.

Figure 4: Visualization examples from the test split of Flickr30k Entities are shown. Solid-lined boxes indicate predicted phrase localizations and dashed-line boxes indicate the ground-truth. If there are multiple ground-truth boxes, the closest box is shown to investigate. Each color of a phrase is matched with the corresponding color of predicted and ground-truth boxes. Best view in color.

### 6.3 Qualitative analysis

The visualization for a two-glimpse BAN is shown in Figure 3. The question is "what color are the pants of the guy skateboarding". The question and content words, *what*, *pants*, *guy*, and *skateboarding* and skateboarder's pants in the image are attended. Notice that the box 2 (orange) captured the sitting man's pants in the bottom.

## 7  Flickr30k entities results and discussions

To examine the capability of bilinear attention map to capture vision-language interactions, we conduct experiments on Flickr30k Entities [23]. Our experiments show that BAN outperforms the previous state-of-the-art on the phrase localization task with a large margin of 4.48% at a high speed of inference.

**Performance.** In Table 4, we compare with other previous approaches. Our bilinear attention map to predict the boxes for the phrase entities in a sentence achieves new state-of-the-art with 69.69% for Recall@1. This result is remarkable considering that BAN does not use any additional features like box size, color, segmentation, or pose-estimation [23, 37]. Note that both Query-Adaptive RCNN [10] and our off-the-shelf object detector [2] are based on Faster RCNN [25] and pre-trained on Visual Genome [17]. Compared to Query-Adaptive RCNN, the parameters of our object detector are fixed and only used to extract 10-100 visual features and the corresponding box proposals.

**Type.** In Table 6 (included in Appendix), we report the results for each type of Flickr30k Entities. Notice that *clothing* and *body parts* are significantly improved to 74.95% and 47.23%, respectively.

**Speed.** The faster inference is achieved taking advantage of multi-channel inputs in our BAN. Unlike previous methods, BAN ables to infer multiple entities in a sentence which can be prepared as a

Table 3: Test-dev and test-standard scores of single-model on VQA 2.0 dataset to compare state-of-the-arts, trained on training and validation splits, and Visual Genome for feature extraction or data augmentation. † This model can be found in `https://github.com/yuzcccc/vqa-mfb`, which is not published in the paper.

| Model | Overall | Yes/no | Number | Other | Test-std |
|---|---|---|---|---|---|
| Bottom-Up [2, 29] | 65.32 | 81.82 | 44.21 | 56.05 | 65.67 |
| MFH [39] | 66.12 | - | - | - | - |
| Counter [41] | 68.09 | 83.14 | 51.62 | 58.97 | 68.41 |
| MFH+Bottom-Up [39]† | 68.76 | 84.27 | 49.56 | 59.89 | - |
| BAN (ours) | 69.52 | 85.31 | 50.93 | 60.26 | - |
| BAN+Glove (ours) | 69.66 | **85.46** | 50.66 | 60.50 | - |
| BAN+Glove+Counter (ours) | **70.04** | 85.42 | **54.04** | **60.52** | **70.35** |

Table 4: Test split results for Flickr30k Entities. We report the average performance of our three randomly-initialized models (the standard deviation of R@1 is 0.17). *Upper Bound* of performance asserted by object detector is shown. † box size and color information are used as additional features. ‡ semantic segmentation, object detection, and pose-estimation is used as additional features. Notice that the detectors of Hinami and Satoh [10] and ours [2] are based on Faster RCNN [25], pre-trained using Visual Genome dataset [17].

| Model | Detector | R@1 | R@5 | R@10 | Upper Bound |
|---|---|---|---|---|---|
| Zhang et al. [40] | MCG [3] | 28.5 | 52.7 | 61.3 | - |
| Hu et al. [11] | Edge Boxes [42] | 27.8 | - | 62.9 | 76.9 |
| Rohrbach et al. [26] | Fast RCNN [7] | 42.43 | - | - | 77.90 |
| Wang et al. [33] | Fast RCNN [7] | 42.08 | - | - | 76.91 |
| Wang et al. [32] | Fast RCNN [7] | 43.89 | 64.46 | 68.66 | 76.91 |
| Rohrbach et al. [26] | Fast RCNN [7] | 48.38 | - | - | 77.90 |
| Fukui et al. [6] | Fast RCNN [7] | 48.69 | - | - | - |
| Plummer et al. [23] | Fast RCNN [7]† | 50.89 | 71.09 | 75.73 | 85.12 |
| Yeh et al. [37] | YOLOv2 [24]‡ | 53.97 | - | - | - |
| Hinami and Satoh [10] | Query-Adaptive RCNN [10] | 65.21 | - | - | - |
| BAN (ours) | Bottom-Up [2] | **69.69** | **84.22** | **86.35** | 87.45 |

multi-channel input. Therefore, the number of forwardings to infer is significantly decreased. In our experiment, BAN takes 0.67 ms/entity whereas the setting that single entity as an example takes 0.84 ms/entity, achieving 25.37% improvement. We emphasize that this property is a novel in our model that considers every interaction among vision-language multi-channel inputs.

**Visualization.** Figure 4 shows the examples from the test split of Flickr30k Entities. The entities which have visual properties, *i.e.*, *a yellow tennis suit* and *white tennis shoes* in Figure 4a, and *a denim shirt* in Figure 4b, are correct. However, a relatively small object (*e.g.*, *a cigarette* in Figure 4b) and the entity that requires semantic inference (*e.g.*, *a male conductor* in Figure 4c) are incorrect.

# 8 Conclusions

BAN gracefully extends unitary attention networks exploiting bilinear attention maps, where the joint representations of multimodal multi-channel inputs are extracted using low-rank bilinear pooling. Although BAN considers every pair of multimodal input channels, the computational cost remains in the same magnitude, since BAN consists of matrix chain multiplication for efficient computation. The proposed residual learning of attention efficiently uses up to eight bilinear attention maps, keeping the size of intermediate features constant. We believe our BAN gives a new opportunity to learn the richer joint representation for multimodal multi-channel inputs, which appear in many real-world problems.

**Acknowledgments**

We would like to thank Kyoung-Woon On, Bohyung Han, Hyeonwoo Noh, Sungeun Hong, Jaesun Park, and Yongseok Choi for helpful comments and discussion. Jin-Hwa Kim was supported by 2017 Google Ph.D. Fellowship in Machine Learning and Ph.D. Completion Scholarship from College of Humanities, Seoul National University. This work was funded by the Korea government (IITP-2017-0-01772-VTT, IITP-R0126-16-1072-SW.StarLab, 2018-0-00622-RMI, KEIT-10060086-RISF). The part of computing resources used in this study was generously shared by Standigm Inc.

## Footnotes

*This work was done while at Seoul National University.

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
