[Supplementary Material]

# Bilinear Attention Networks — Appendix

## A    Variants of BAN

### A.1    Enhancing glove word embedding

We augment a computed 300-dimensional word embedding to each 300-dimensional Glove word embedding. The computation is as follows: 1) we choose arbitrary two words $w_i$ and $w_j$ from each question that can be found in VQA and Visual Genome datasets or each caption in MS COCO dataset. 2) we increase the value of $\mathbf{A}_{i,j}$ by one where $\mathbf{A} \in \mathbb{R}^{V' \times V'}$ is an association matrix initialized with zeros. Notice that $i$ and $j$ can be the index out of vocabulary $V$ and the size of vocabulary in this computation is denoted by $V'$. 3) to penalize highly frequent words, each row of $\mathbf{A}$ is divided by the number of sentences (question or caption) which contain the corresponding word. 4) each row is normalized by the sum of all elements of each row. 5) we calculate $\mathbf{W}' = \mathbf{A} \cdot \mathbf{W}$ where $\mathbf{W} \in \mathbb{R}^{V' \times E}$ is a Glove word embedding matrix and $E$ is the size of word embedding, *i.e.*, 300. Therefore, $\mathbf{W}' \in \mathbb{R}^{V' \times E}$ stands for the mixed word embeddings of semantically closing words. 6) finally, we select $V$ rows from $\mathbf{W}'$ corresponding to the vocabulary in our model and augment these rows to the previous word embeddings, which makes 600-dimensional word embeddings in total. The input size of GRU is increased to 600 to match with these word embeddings. These word embeddings are fine-tuned.

As a result, this variant significantly improves the performance to 66.03 ($\pm 0.12$) compared with the performance of 65.72 ($\pm$ 0.11) which is done by augmenting the same 300-dimensional Glove word embeddings (so the number of parameters is controlled). In this experiment, we use four-glimpse BAN and evaluate on validation split. The standard deviation is calculated by three random initialized models and the means are reported. The result on test-dev split can be found in Table 3 as *BAN+Glove*.

### A.2    Integrating counting module

The counting module [41] is proposed to improve the performance related to counting tasks. This module is a neural network component to get a dense representation from the spatial information of detected objects, *i.e.*, the left-top and right-bottom positions of the $\phi$ proposed objects (rectangles) denoted by $\mathbf{S} \in \mathbb{R}^{4 \times \phi}$. The interface of the counting module is defined as:

$$\mathbf{c} = \text{Counter}(\mathbf{s}, \tilde{\alpha}) \tag{14}$$

where $\mathbf{c} \in \mathbb{R}^{\phi+1}$ and $\tilde{\alpha} \in \mathbb{R}^{\phi}$ is the logits of corresponding objects for $\text{sigmoid}$ function inside the counting module. We found that the $\tilde{\alpha}$ defined by $\max_{j=1,...,\phi}(A_{\cdot,j})$, *i.e.*, the maximum values in each column vector of $A$ in Equation 9, was better than that of summation. Since the counting module does not support variable-object inputs, we select 10-top objects for the input instead of $\phi$ objects based on the values of $\tilde{\alpha}$.

The BAN integrated with the counting module is defined as:

$$\mathbf{f}_{i+1} = \left(\text{BAN}_i(\mathbf{f}_i, \mathbf{Y}; \mathcal{A}_i) + g_i(\mathbf{c}_i)\right) \cdot \mathbb{1}^T + \mathbf{f}_i \tag{15}$$

where the function $g_i(\cdot)$ is the i-th linear embedding followed by ReLU activation function and $\mathbf{c}_i = \text{Counter}(\mathbf{s}, \max_{j=1,...,\phi}(A_{\cdot,j}^{(i)}))$ where $A^{(i)}$ is the logit of $\mathcal{A}_i$. Note that a dropout layer before this linear embedding severely hurts performance, so we did not use it.

As a result, this variant significantly improves the counting performance from 54.92 ($\pm 0.30$) to 58.21 ($\pm 0.49$), while overall performance is improved from 65.81 ($\pm 0.09$) to 66.01 ($\pm 0.14$) in a controlled experiment using a vanilla four-glimpse BAN. The definition of a subset of counting questions comes from the previous work [30]. The result on test-dev split can be found in Table 3 as *BAN+Glove+Counter*, notice that, which is applied by the previous embedding variant, too.

## A.3 Integrating multimodal factorized bilinear (MFB) pooling

Yu et al. [39] extend low-rank bilinear pooling [15] with the rank $\mathbf{k} > 1$ and two factorized three-dimensional matrices, which called as MFB. The implementation of MFB is effectively equivalent to low-rank bilinear pooling with the rank $d' = d \times \mathbf{k}$ followed by sum pooling with the window size of $\mathbf{k}$ and the stride of $\mathbf{k}$, defined by $\mathrm{SumPool}(\tilde{\mathbf{U}}^T \mathbf{x} \circ \tilde{\mathbf{V}}^T \mathbf{y}, \mathbf{k})$. Notice that a pooling matrix $\mathbf{P}$ in Equation 2 is not used. The variant of BAN inspired by MFB is defined as:

$$\mathbf{z}_{k'} = \sigma(\mathbf{X}^T \tilde{\mathbf{U}})_{k'}^T \cdot \mathcal{A} \cdot \sigma(\mathbf{Y}^T \tilde{\mathbf{V}})_{k'} \tag{16}$$

$$\mathbf{f}' = \mathrm{SumPool}(\mathbf{z}, \mathbf{k}) \tag{17}$$

where $\tilde{\mathbf{U}} \in \mathbb{R}^{N \times K'}$, $\tilde{\mathbf{V}} \in \mathbb{R}^{M \times K'}$, $\sigma$ denotes $\mathrm{ReLU}$ activation function, and $\mathbf{k} = 5$ following Yu et al. [39]. Notice that $K' = K \times \mathbf{k}$ and $k'$ is the index for the elements in $\mathbf{z} \in \mathbb{R}^{K'}$ in our notation.

However, this generalization was not effective for BAN. In Figure 2b, the performance of *BAN-1+MFB* is not significantly different from that of *BAN-1*. Furthermore, the larger $K'$ increases the peak consumption of GPU memory which hinders to use multiple-glimpses for the BAN.

Table 5: Test-standard scores of ensemble-model on VQA 2.0 dataset to compare state-of-the-arts. Excerpt from the VQA 2.0 Leaderboard at the time of writing. # denotes the number of models for their ensemble methods.

| Team Name | # | Overall | Yes/no | Number | Other |
|---|---|---|---|---|---|
| vqateam_mcb_benchmark [6, 8] | 1 | 62.27 | 78.82 | 38.28 | 53.36 |
| vqa_hack3r | - | 62.89 | 79.88 | 38.95 | 53.58 |
| VQAMachine [34] | - | 62.97 | 79.82 | 40.91 | 53.35 |
| NWPU_VQA | - | 63.00 | 80.38 | 40.32 | 53.07 |
| yahia zakaria | - | 63.57 | 79.77 | 40.53 | 54.75 |
| ReasonNet_ | - | 64.61 | 78.86 | 41.98 | 57.39 |
| JuneflowerIvaNlpr | - | 65.70 | 81.09 | 41.56 | 57.83 |
| UPMC-LIP6 [4] | - | 65.71 | 82.07 | 41.06 | 57.12 |
| Athena | - | 66.67 | 82.88 | 43.17 | 57.95 |
| Adelaide-Teney | - | 66.73 | 83.71 | 43.77 | 57.20 |
| LV_NUS [12] | - | 66.77 | 81.89 | 46.29 | 58.30 |
| vqahhi_drau | - | 66.85 | 83.35 | 44.37 | 57.63 |
| CFM-UESTC | - | 67.02 | 83.69 | 45.17 | 57.52 |
| VLC Southampton [41] | 1 | 68.41 | 83.56 | 51.39 | 59.11 |
| Tohoku CV | - | 68.91 | 85.54 | 49.00 | 58.99 |
| VQA-E | - | 69.44 | 85.74 | 48.18 | 60.12 |
| Adelaide-Teney ACRV MSR [29] | 30 | 70.34 | 86.60 | 48.64 | 61.15 |
| DeepSearch | - | 70.40 | 86.21 | 48.82 | 61.58 |
| HDU-USYD-UNCC [39] | 8 | 70.92 | 86.65 | 51.13 | 61.75 |
| BAN+Glove+Counter (ours) | 1 | 70.35 | 85.82 | 53.71 | 60.69 |
| BAN Ensemble (ours) | 8 | 71.72 | 87.02 | **54.41** | 62.37 |
| BAN Ensemble (ours) | 15 | **71.84** | **87.22** | 54.37 | **62.45** |

Table 6: Recall@1 performance over types for Flickr30k Entities (%)

| Model | People | Clothing | Body Parts | Animals | Vehicles | Instruments | Scene | Other |
|---|---|---|---|---|---|---|---|---|
| Rohrbach et al. [26] | 60.24 | 39.16 | 14.34 | 64.48 | 67.50 | 38.27 | 59.17 | 30.56 |
| Plummer et al. [23] | 64.73 | 46.88 | 17.21 | 65.83 | 68.75 | 37.65 | 51.39 | 31.77 |
| Yeh et al. [37] | 68.71 | 46.83 | 19.50 | 70.07 | 73.75 | 39.50 | 60.38 | 32.45 |
| Hinami and Satoh [10] | 78.17 | 61.99 | 35.25 | 74.41 | 76.16 | **56.69** | 68.07 | 47.42 |
| BAN (ours) | **79.90** | **74.95** | **47.23** | **81.85** | **76.92** | 43.00 | **68.69** | **51.33** |
| # of Instances | 5,656 | 2,306 | 523 | 518 | 400 | 162 | 1,619 | 3,374 |