[Reviews · NeurIPS 2018]

Reviewer 1



Update after Author Feedback: Thanks for addressing my ask for empirical speed tradeoff, clarifying the answer space, and for open sourcing the code. Nice work! --------------------------- Original Review: This paper proposes a new attention mechanism when the model has two different kinds of inputs that can be considered to have multiple channels. Similar to other forms of co-attention or bilinear attention, these bilinear attention networks attempt to combine information from both inputs into a single representation. Though the exposition isa bit dense at times, the BANs appear to provide a simple, effective way to boost performance on tasks like VQA and Flicker30kEntities to new state-of-the-art performances. The idea in this paper is relatively simple despite the many lines of equations; the insights offered by additional experiments in Figure 2 are useful as well, and because the models seem so simple, there is little additional ablation I found wanting. Overall, this leads to a 7 for strong empirical results with a simple idea, with various different options and settings explored for two difficult multimodal datasets. Confidence here is a 3 because I lack the familiarity with VQA and Flickr30kEntities to gauge whether this approach is too similar to or incremental compared to other recent approaches. I can tell that this is a simple, effective method that builds off of the low-rank bilinear models of Pirsiavash, bilinear pooling, and multi-head attention, but I may have gaps in my knowledge to tell whether this is truly novel in that context. Math and other details were carefully checked though. I would have preferred to see some experiments demonstrating the speed tradeoff of using BANs empirically (rather than just theoretically) to put that more into perspective. line 165-166 “The number of outputs is determined by the 
minimum occurrence of the answer in unique questions as nine times in the dataset, which is 3,129.” Can you clarify what this means? I can’t tell what ‘as nine times in the dataset’ means. I’m currently interpreting it as you are allowing the output of the model to be 3129 different answers, which are the answers that occur in the dataset at least 9 times. I'm reserving very confident for reproducibility of systems that have been declared as being open source in the future. This submission includes a lot of detail about hyperparameters and training. The model is simple enough it would seem. But nonetheless, without open sourced code people might run into problems and unanswered questions during reimplementation.

Reviewer 2



This paper proposes bilinear attention networks which extends co-attention network to consider every pair of multimodal channels. The authors propose bilear attention networks to enable bilinear attention distribution atop low-rank bilinear pooling. Besides, the paper also introduces a variant of multimodal residual network to efficiently integrate the joint representations from the multiple bilinear attention maps. By doing this, in the experiments, the bilinear attention network can allow up to 8-head attention, and shows the effectiveness of the network on both Visual QA task and visual grounding by comparing with some state-of-the-art approaches. The paper seems cite a good amount of recent significant publications on attention networks and bilinear pooling. The paper is in general well-written and easy to follow. Please find some of the minor suggestions or issues as below: 1) In the experiments, what the impact of adding the nonlinearity is? It would be good if incorporating the discussion or result in the paper. 2) Instead of using learning rate decay, two alternatives would be either using NT-SGD : Polyak, B. and Juditsky, A. Acceleration of stochastic approximation by averaging. SIAM Journal on Control and Optimization, 30(4):838–855, 1992. or using a learning scheduler (similarity based etc.). 3) In Table 2, it would be good if using nparams=1280 to be the same as for co-attention for BAN to show the improvements. 4) It will be also helpful to show the running time performance compared with co-attention. 5) typo: in the BAN section, the j-th channel (*column*) of input Y.

Reviewer 3



The main contribution of this paper is a multimodal attention that links previous work of low-rank pooling, with Co-Attention methodologies, i.e., use of affinity matrix between modalities. Being competitive, the paper combines of many of the recent advancement, such as Down/Up attention, glimpses and counter model. Down/Up methodology with Co-Attention is new, and beautifully illustrated, showing an exact interaction between question elements and detected-object regions. The final model achieve SOTA results in for a very competitive task. Clarity/Quality: The paper is well-written. contributions clearly laid out. Originality: The paper inspired and closely follows work of Lu et. al. [18] parallel model combines with low-rank bilinear pool, such as MCB, MLB, MFH. Significance: The proposed approach demonstrates clear improvements for VQA task. Analysis of different attention methods proves the proposed attention is better than previous work. Concerns: •Related work of the paper is not accurate. L123-124 claim that Co-Attention neglected interactions between modalities. The parallel version of [18] or [20] defiantly capture the interactions between the modalities via similarity matrix. The parallel version was ignored in Quantitative analysis. I think you should elaborate about the exact differences between your technique, and parallel-like techniques. •Related work is also not comprehensive enough in my opinion. As attention paper, the discussion is limited only to very similar approaches. For instance, “Structured Attentions for Visual Question Answering” from ICCV17, solves multimodal attention as structure prediction, or ‘’High-Order Attention Models for Visual Question Answering from NIPS17, which also discuss Co-Attention, and suggests a network with combined unary and pairwise information (i.e., multiply interactions), which surpass [18], and was not mentioned. •Fig 2. How exactly have you tested the different attentions? Did you replace only the attention unit? Did you tune the network with different architecture? • why this model is avoiding overfitting, or able to use more glimpses? Conclude: I like the paper, though it feels a bit too concentrated on the competitive aspects, i.e. accuracy score. I am missing a more comprehensive justification and comparison to similar co-attention techniques. The attention unit is novel, but incremental from common co-attention practice. Nevertheless, the overall quality of the submission is good, with SOTA performance and a complete analysis for many design choices, therefore I’m in favor for acceptance. After author response: I thank the authors for their response. I never intended to misjudge the significance of BAN results vs previous work, I was more confused about the important ideas of BAN versus other approaches that use bilinear attention. While the difference from iteratively approaches for co-attention is obvious, I'm still confused about the difference from previous work that use bilinear attention. The paper only mention a very early approach by Xu and Saenko. The proposed low rank bilinear operation is not new, and was already suggested in the papers I mentioned. Arguing that bilinear approaches are weak against iterative approaches does not align with the paper main argument. To progress the research in this direction, I hope you will chose to clarify the importance of your approach for attention in the revised version, and looking forward to see the paper published at NIPS.